# Effect of a decision aid on the choice of pregnant women whether to have epidural anesthesia or not during labor

Eri Shishido[1]* , Wakako Osaka[2‡], Ayame Henna[3‡], Yuko Motomura[4‡], Shigeko Horiuchi[1]

**1** St. Luke's International University, Tokyo, Japan, **2** Keio University, Tokyo, Japan, **3** St. Luke's International Hospital, Tokyo, Japan, **4** Tokyo-Kita Medical Center, Tokyo, Japan

☯ These authors contributed equally to this work.
‡ This author also contributed equally to this work.
* eri-shishido@slcn.ac.jp

## Abstract

### Objective

Decision aids (DAs) are useful in providing information for decision-making on using epidural anesthesia during birth. To date, there has been little development of DAs for Japanese pregnant women. Herein, we investigated the effect of a DA on the decision of pregnant women whether to have epidural anesthesia or not for labor during vaginal delivery. The primary outcome was changes in mean decision conflict score.

### Methods

In this non-randomized controlled trial, 300 low-risk pregnant women in an urban hospital were recruited by purposive sampling and assigned to 2 groups: DA (intervention) and pamphlet (control) groups. Control enrollment was started first (until 150 women), followed by intervention enrollment (150 women). Pre-test and post-test scores were evaluated using the Decision Conflict Scale (DCS) for primary outcome, knowledge of epidural anesthesia and satisfaction with decision making for secondary outcomes, and decision of anesthesia usage (i.e., with epidural anesthesia, without epidural anesthesia, or undecided).

### Results

Women in the DA group (n = 149: 1 excluded because she did not return post-test questionnaire) had significantly lower DCS score than those in the pamphlet group (n = 150) (DA: -8.41 [SD 8.79] vs. pamphlet: -1.69 [SD 5.91], p < .001). Knowledge of epidural anesthesia and satisfaction with decision-making scores of women who used the DA were significantly higher than those of women who used the pamphlet (p < .001). Women in the DA group showed a significantly lower undecided rate than those in the pamphlet group. The number of undecided women in the DA group significantly decreased from 30.2% to 6.1% (p < .001), whereas that in the pamphlet group remained largely unchanged from 40.7% to 38.9%.

**Data Availability Statement:** All relevant data are within the manuscript and its Supporting Information files.

**Funding:** This study was supported by grants from the Japan Society for the Promotion of Science (P1-Shigeko Horiuchi 17H01613) and (P2-Naoko Arimori 17H04427) (https://www.jsps.go.jp/english/e-grants/grants01.html).

**Competing interests:** The authors have declared that no competing interests exist.

## Conclusion

This study indicates that a DA can be useful in helping women make a decision whether to have epidural anesthesia or not for labor during vaginal delivery.

## Introduction

In Japan, the proportion of women receiving epidural anesthesia increased from 2.6% in 2008 to 6.2% in 2016 [1, 2]. However, compared with other countries, the rate of analgesic delivery in Japan is low. This is because of the small percentage of hospitals that can provide epidural anesthesia (15.0%, 2017), failure to provide information about epidural anesthesia to all pregnant women, and a cultural belief that labor pain is a virtue [3, 4]. In Japan, about half of deliveries are performed in hospitals and clinics. As there are no anesthesiologists in all facilities, only a few are able to perform epidural anesthesia. Although the majority of Japanese women wish to participate in making decisions regarding their healthcare [5, 6], little is known regarding the decision-making needs or conflicts of Japanese women facing labor pain.

Another study in Japan [7] examined groups of women who gave birth in accordance with their decision to use epidural anesthesia. One group consisted of 271 (44.5%) women who had epidural anesthesia. Another group was composed of 239 (39.2%) women who did not have epidural anesthesia. There was also a group made up of 99 (16.3%) women who were undecided, and half of these women received epidural anesthesia. These women who were undecided had a lower mean age and consisted of more primiparas than the women who chose to have or not to have epidural anesthesia. The study suggested that midwives need to provide sufficient information to help women make the appropriate decision regarding their childbirth.

A study in Japan has noted that midwives have also provided the same level of information as obstetricians in the maternity class. This suggests that decision-making support during the perinatal period should be provided equally by midwives and obstetricians [4]. The anesthesiologist provides information on the indications, contraindications, and benefits/risks of anesthesia in the outpatient setting. In this context, shared decision-making (SDM) may be considered as an effective method to providing childbirth information.

SDM between a healthcare provider and a patient has been reported to be a successful method for helping patients make their decisions [8]. In this regard, Decision Aids (DAs) have been carefully developed as educational tools to enhance a patient's ability to choose from among various treatment options. These DAs may be in the form of pamphlets, websites, or videos to support choices that are consistent with the patient's preferences and values [9]. In their systematic review, Stacey et al. reported that DAs reduced decision conflicts, increased decision satisfaction and knowledge, and decreased the proportion of participants who were unable to make decisions [10].

It is therefore logical to presume that DAs would be useful in providing information for decision-making on the use of epidural anesthesia during birth. Unfortunately, there has been little development of DAs for pregnant women to date. In 2010, Raynes-Greenow et al. [11] developed a DA for choosing a pain relief method for labor, and whose effectiveness was evaluated in a randomized controlled trial (RCT). The DA included systematic review sources for each of the 10 identified pain relief methods available to pregnant women. However, because of the many options of labor pain relief methods, neither of these methods resulted in a reduction in decision conflict. The present study differs from the study of Raynes-Greenow et al.

[11] in that it investigated the decision-making of pregnant women with regard to the use of analgesia during birth. Specifically, this study aimed to investigate the effect of a DA on pregnant women's decision whether to use epidural anesthesia or not when planning to have vaginal delivery. The primary outcome was changes in the mean decision conflict score.

## Hypotheses

Our main hypothesis is that the utilization of a DA as intervention is useful in helping prenatal women decide on the use of epidural anesthesia when planning for vaginal delivery. Our specific hypotheses are as follows:

1. The mean decision conflict score is reduced in the intervention group compared with the control group.

2. The mean satisfaction with decision-making score after the intervention is higher in the intervention group than in the control group.

3. The mean knowledge score of epidural anesthesia is higher in the intervention group than in the control group.

4. The proportion of pregnant women who are undecided whether to deliver with or without epidural anesthesia is decreased in the intervention group compared with the control group.

## Materials and methods

### Study design and participants

In this non-randomized (non-equivalent) controlled trial, the participants included low-risk pregnant women of singleton pregnancies who (1) were planning to have vaginal delivery, (2) were between 34 and 36 gestation weeks before visiting the obstetric anesthesiology unit, and (3) could communicate, read, and write in Japanese. The exclusion criteria included pregnant women with (1) an obstetric complications history (e.g., cesarean section), (2) a medical history (e.g., mental disorder), (3) a fetal disease, and (4) contraindications (e.g., blood coagulation disorders, aortic stenosis, and hypertrophic cardiomyopathy).

**Sample size.**   Stacey et al. and Raynes-Greenow reported that the intervention is effective if there is a difference in the conflict scores of -4.35 (95% CI -6.8 to -1.9) [10, 11] with a calculated 141 people needed in each group. In the present study, we also considered a dropout rate of about 10% and thus set the number of participants to 150 in each group for a total of 300 participants recruited (effect size 0.5; power 0.8; significance level 5%).

**Study protocol.**   Participants were selected using purposive sampling. Enrollment for the control group was started first and this continued until 150 women who met the inclusion criteria agreed to participate and whose data were collected. Recruitment, enrollment, and data collection for the intervention group were conducted after the recruitment for the control group and followed the same protocol. The researcher (ES) who is a certified midwife at a hospital in Tokyo, Japan conducted the study. Data allocation for the control group was performed first from June 1 to July 10, 2019 (**Fig 1**). Data allocation for the intervention group was performed from July 15 to August 27, 2019. Data were collected from June 2019 to the end of August 2019. The Ethics Board of St. Luke's International University approved this study (18-A069). The hospital has about 2,300 deliveries per year, and about 60% of these deliveries choose to have an epidural anesthesia. The hospital can perform an epidural anesthesia any time.

| | pre-test | 1-2 weeks | post-test | pre-test | 1-2 weeks | post-test |
|---|---|---|---|---|---|---|
| Control group | × | Pamphlet | × | | | |
| Intervention group | | | | × | DA | × |

（×）Questionnaire: DCS, Satisfaction with decision-making, Knowledge of epidural anesthesia, Choice of epidural anesthesia

**Fig 1. Timeline of intervention procedure.**

When eligible pregnant women at 34 weeks of gestation visited for their maternity check-up, the researcher (ES) verbally explained the study using the explanatory booklet containing information about the provision of confidentiality and anonymity of their data. The pre-test and post-test were consecutively numbered and anonymously consolidated. The questionnaire was given by the researcher (ES) and the pre-test asked the participants to indicate their consent to the study. A check mark in the consent box was considered to indicate voluntary participation in the study. A pre-test consisting of 4 measurement tools was administered to all eligible women who had agreed to participate during their antenatal check-up after 34 weeks of gestation.

Participants in the control group received only the standard information pamphlet as the usual care from the researcher (ES) after the pre-test. The pamphlet, which consists of 10 pages of A4-sized paper, provides information on the benefits and risks of epidural anesthesia.

Participants in the intervention group received the DA which consists of 22 pages of A4-sized paper from the researcher (ES). It took about 20 minutes for the participants to read the DA or pamphlet.

The DA (for the intervention group) and standard information pamphlet (for the control group) showed differences in the following items: 1) information on epidural anesthesia or no epidural anesthesia options, 2) comparative tables of each option, 3) values clarification exercise, and 4) decision-making process. The pamphlet only described in writing the benefits and risks of epidural anesthesia. The content is purely informational and does not include a decision-making component found in the DA.

These DA and pamphlet were to be read before the next antenatal check-up. At the next check-up about a week later, after confirming that the participants had read the pamphlet or DA, a post-test containing the same measurements was conducted. At the end of the post-test, the intervention group was given the pamphlet and the control group was given the DA.

## Decision aid development

To adhere to the systematic development of DAs, we followed the guidelines of Coulter et al. (2013) [12]. They indicated that to complete the final DA, a group of patients and clinicians must first be identified by selecting the patients and determining their objectives. In the present study, we accomplished patient selection and objective determination using outcomes from previous studies [4, 7, 13]. Then, we identified the individual needs of the patients [7] and clinicians, and integrated them. The needs of the patients were knowledge of the analgesic and how to participate in making decisions. The need of the clinicians was to understand the patient's preference. As Coulter et al. (2013) [12] recommended, the needs of the patients and clinicians were integrated with reviews and evidence.

Next, we searched PubMed, Cochrane Database of Systematic Reviews, and Japan Medical Abstracts Society (version 5) for reports related to our study with restrictions from January

2003 to May 2018. We used the keywords "decision-making", "decision aid", "epidural analgesia", "pain relief", and "labor". Of 179 reports initially identified, 8 trials were included in the final review and used to determine the contents of the DA prototype.

We used the Ottawa Patient Decision Aid Development eTraining [14] as a reference for the development of the DA prototype. We selected the Ottawa Decision Support Framework [15] as the support medium and intervention method. The specific items of the DA prototype were adapted from the decision support aids on painless labor developed by the Healthwise Content Development Team (2018) [16] and the aids on labor pain relief produced by the University of Sydney (2004) [17]. In addition, the structure of the DA prototype was based on the decision-making guide for breast cancer surgery methods by Osaka et al. (2014) [18]. It was designed to follow the decision-making process in the following order: STEP 1: Know how to make a decision with conviction; STEP 2: Understand the characteristics of the options; STEP 3: Clarify what is important to you; STEP 4: Make the decision.

The contents of the DA prototype were listed to make it easier to compare the benefits and risks between no analgesia and epidural analgesia. Risks posed by each method were presented in pictograms and a circle graph for clarity.

Coulter et al. (2013) [12] recommended that a DA prototype must undergo alpha testing by a group of experts to review, evaluate, and modify its content. We completed the alpha testing by consulting with a perinatal physician (n = 1), anesthesiologist (n = 2), experienced midwives (n = 3), experts in decision-making (n = 1), and women (n = 15) with childbirth experience. Subsequently, we conducted beta testing to ascertain feasibility based on acceptability. Fifteen mothers provided data about the acceptability of the DA prototype which were used for making changes in the content for additional clarity and face validity.

The development process and shared decision-making design components of the DA prototype were evaluated using the Japanese version of the International Patient Decision Aid Standards instrument (IPDASi version 4.0). The DA prototype met many quality standards of the Japanese version of IPDASi (version 4.0) developed by Osaka et al. (2017) [19]. Specifically, the DA prototype met all 6 *qualifying criteria* of the Japanese version of IPDASi (version 4.0) to be considered as a DA, that is, 6 out of 6 certification criteria after excluding 4 items that were not related to the examination for a high risk of harmful bias, and 14 out of 28 *quality criteria* considered to strengthen a DA but whose omission does not present a high risk of harmful bias. These 6 qualifying criteria should all be met to be classified as a DA, and meeting all the certification criteria indicates no harmful bias. Therefore, the quality of the DA is ensured.

## Measurements

**Primary outcome: Decisional conflict.** The Decision Conflict Scale (DCS) is a 16-item self-report questionnaire that measures patient's uncertainty about which treatment to choose, factors contributing to uncertainty (e.g., believing oneself to be uninformed, unclear values, unsupported in decision-making), and perceived effectiveness of decision-making [20]. The DCS has 5 subscales: Informed, Clarity, Support, Uncertainty, and Effective decision. Questions must be answered using a 5-point Likert scale (0 = strongly agree, 4 = strongly disagree). Decisional conflict scores were calculated according to the DCS user manual and can range from 0 to 100 [20]. The lower the score the lower the decisional conflict. In the present study, we evaluated decision conflict using the Japanese version of DCS created by Kawaguchi et al. (2013) [21]. This Japanese version demonstrated high internal consistency (Cronbach's alpha = 0.84 − 0.96) [21].

**Secondary outcomes: Satisfaction with decision-making, knowledge of epidural anesthesia, and choice of epidural anesthesia.** The participant's satisfaction with decision-

making was assessed using the effective decision-making subscale of the DCS. The subscale consists of 4 items on satisfaction with decision-making. Scores for the participant's satisfaction with decision-making are reversed, thus higher scores reflect higher levels of satisfaction. Satisfaction with decision-making scores was calculated in the same manner as the decisional conflict scores and can range from 0 to 100.

We created a general knowledge test according to the questions of Raynes-Greenow et al. (2010) [11] including the advantages and disadvantages of epidural anesthesia. Ten questions were answerable with "yes" or "no". The highest score was 10 points, indicating a good source of knowledge about epidural anesthesia. The decision on using anesthesia for labor during vaginal delivery (i.e., with epidural anesthesia, without epidural anesthesia, or undecided) was determined using a single question.

**Demographic data.** Demographic data were self-reported as part of the questionnaire. Data collected included age, educational level, and parity.

### Data analysis

The data were descriptively analyzed. To determine if the data were normally distributed, a histogram was used to illustrate the distribution. Sample size was calculated from the difference in the mean DCS scores for the primary outcome, and therefore was analyzed by the difference in mean scores. The baseline characteristics and outcomes of the participants were compared between women in the intervention group and women in the control group using the independent t-test or chi-square test. Statistical analyses were performed using IBM SPSS Statistics (version 24.0; Static Base and Advanced Statistics, IBM Japan, Tokyo, Japan). All statistical tests were performed with a two-sided 5% level of significance.

## Results

### Study flow and participant characteristics

Between July 2019 and August 2019, 300 pregnant women who met the eligibility criteria were invited to participate in this study (**Fig 2**). There were 150 women assigned to the control group and 150 women assigned to the intervention group. In the intervention group, 1 woman failed to return the post-test questionnaire so she was excluded from the study. Thus, the final number of women enrolled in the intervention group was 149 (99.7%).

The demographic data of the participants are shown in **Table 1**. The mean age of the participants in the control group was 33.5 years (SD 5.10), and the range was 20–40 years. There was no significant difference between the 2 groups except for the educational levels, which had missing data from 15 participants (4 from intervention group; 11 from control group). For a minority of participants, the highest educational level was junior high and high school (n = 7; 4.7%) in the control group and (n = 0; 0.0%) in the intervention group.

### Primary outcome

The mean score differences in the primary outcomes from pre-test to post-test in the intervention and control groups are shown in **Table 2**. The pre-test showed that the control group had a significantly higher DCS score (0–100) than the intervention group (intervention group: 31.9 [SD 13.5] vs. control group: 36.2 [SD 14.3], p = .009). Women assigned to the intervention group had a significantly lower mean DCS score than women assigned to the control group (intervention group: -8.41 [SD 8.79] vs. control group: -1.69 [SD 5.91], p < .001). The mean score differences in the DCS subscales were significantly lower in the intervention group than in the control group.

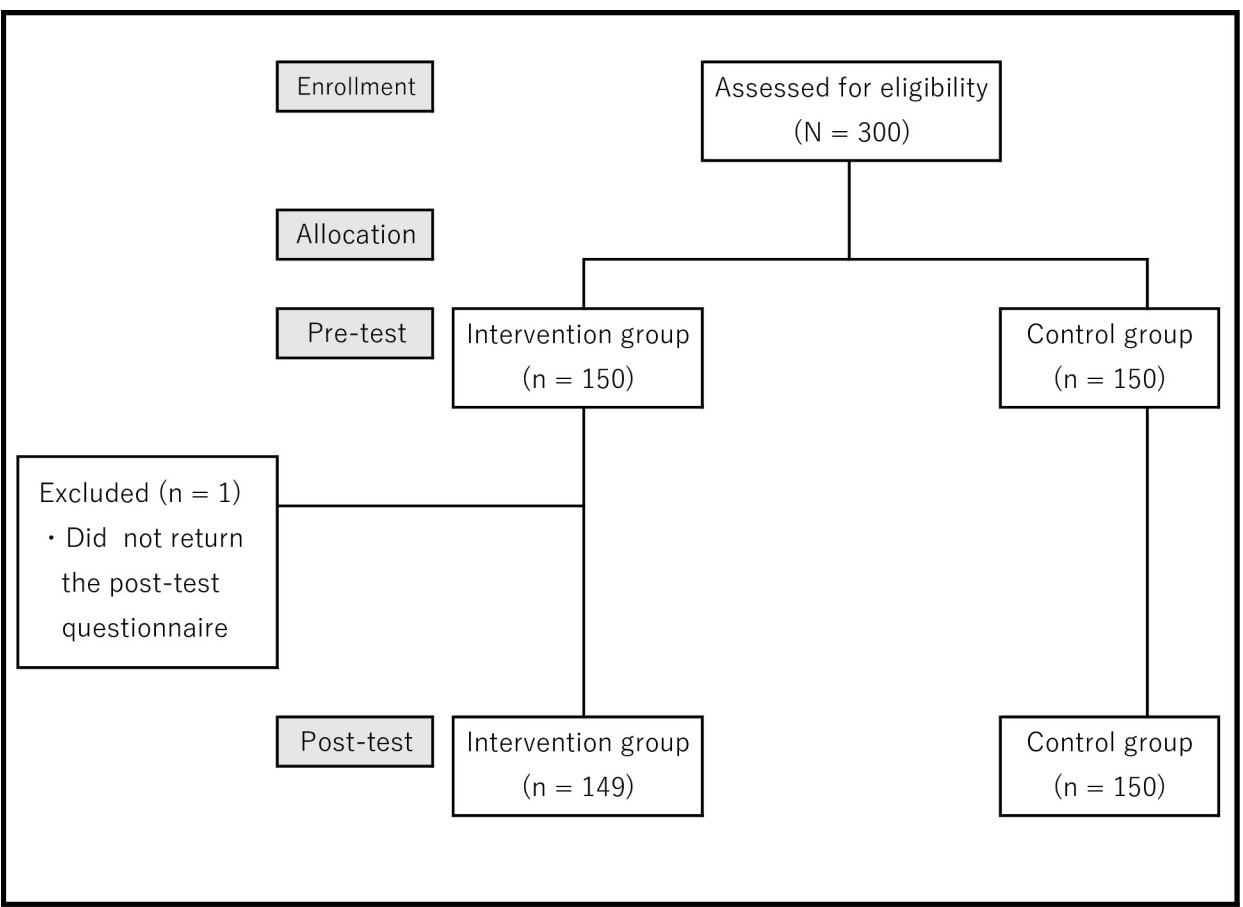

**Fig 2. Participant flow diagram.**

## Secondary outcomes

The mean score differences in the secondary outcomes from pre-test to post-test in the intervention and control groups are shown in **Table 3** and **Fig 3**. The pre-test showed that the intervention group had a significantly higher satisfaction with decision-making score (0–100) than

**Table 1. Demographic data of participants.**

| | Intervention group (n = 149)* | Control group (n = 150) | t | p-value |
|---|---|---|---|---|
| Age (years) | 32.8 (SD 4.72) | 33.5 (SD 5.10) | 1.25 | .214 |
| Educational level[a] | | | | |
| < 12 years | 0 (0.0%) | 7 (4.7%) | | .006 |
| > 12 years | 145 (97.3%) | 132 (88.0) | | |
| Parity | | | | |
| Primipara | 117 (78.5%) | 111 (74.0%) | | .358 |
| Multipara | 32 (21.5%) | 39 (26.0%) | | |

* 1 excluded (did not return post-test questionnaire)

[a]Educational level had missing data from 15 participants (4 from intervention group; 11 from control group)

**Table 2. Mean score differences in the primary outcomes from the pre-test to the post-test in the intervention and control groups.**

| | Intervention group (n = 149)* | Control group (n = 150) | t | p-value |
|---|---|---|---|---|
| **Primary outcomes** | | | | |
| DCS 16 items total | −8.41 (SD 8.79) | −1.69 (SD 5.91) | 7.71 | < .001 |
| pre-test (0–100) | 31.9 (SD 13.5) | 36.2 (SD 14.3) | 2.63 | .009 |
| post-test (0–100) | 23.5 (SD 8.61) | 34.7 (SD 13.8) | 8.40 | < .001 |
| <subscale> | | | | |
| Informed | −12.8 (SD 11.1) | −3.29 (SD 9.58) | 7.94 | < .001 |
| Clarity | −6.93 (SD 11.3) | −2.22 (SD 9.63) | 3.86 | < .001 |
| Support | −6.43 (SD 10.3) | −1.66 (SD 7.78) | 4.51 | < .001 |
| Uncertainty | −5.92 (SD 11.2) | −1.13 (SD 6.87) | 4.41 | < .001 |

* 1 excluded (did not return post-test questionnaire)

Note: DCS = Decision Conflict Scale

the control group (intervention group: 67.2 [SD 15.2] vs. control group: 62.7 [SD 15.1], p = .013). There was a significant difference in the satisfaction with decision-making score between the 2 groups (intervention group: 9.52 [SD 10.9] vs. control group: 0.76 [SD 6.29], p < .001). The pre-test showed that the intervention group had a significantly higher knowledge score (0–10) than the control group (intervention group: 7.34 [SD 1.20] vs. control group: 6.59 [SD 1.33], p < .001). The mean score of knowledge of epidural anesthesia was significantly higher in women assigned to the intervention group than in women assigned to the control group (intervention group: 1.96 [SD 1.63] vs. control group: 1.33 [SD 1.44], p < .001). The number of pregnant women who were undecided whether to deliver with or without epidural anesthesia decreased significantly from 45 (30.2%) to 9 (6.1%) in the intervention group (p < .001) and only from 61 (40.7%) to 58 (38.9%) in the control group.

## Discussion

This study set out to evaluate the effect of a DA on pregnant women's decision on using anesthesia for labor during vaginal delivery in terms of *Decision conflict*, *Knowledge of epidural anesthesia*, *Satisfaction with decision-making*, and *Choice of epidural anesthesia*. Four hypotheses were met in this study.

**Table 3. Mean score differences in the secondary outcomes from the pre-test to the post-test in the intervention and control groups.**

| | Intervention group (n = 149)* | Control group (n = 150) | t | p-value |
|---|---|---|---|---|
| **Secondary outcomes** | | | | |
| Satisfaction with decision-making | 9.52 (SD 10.9) | 0.76 (SD 6.29) | 8.47 | < .001 |
| pre-test (0–100) | 67.2 (SD 15.2) | 62.7 (SD 15.1) | 2.51 | .013 |
| post-test (0–100) | 76.7 (SD 9.72) | 63.6 (SD 14.9) | 8.98 | < .001 |
| Knowledge of epidural anesthesia | 1.96 (SD 1.63) | 1.33 (SD 1.44) | 3.57 | < .001 |
| pre-test (0–10) | 7.34 (SD 1.20) | 6.59 (SD 1.33) | 5.11 | < .001 |
| post-test (0–10) | 9.30 (SD 1.32) | 7.92 (SD 1.50) | 8.46 | < .001 |

* 1 excluded (did not return post-test questionnaire)

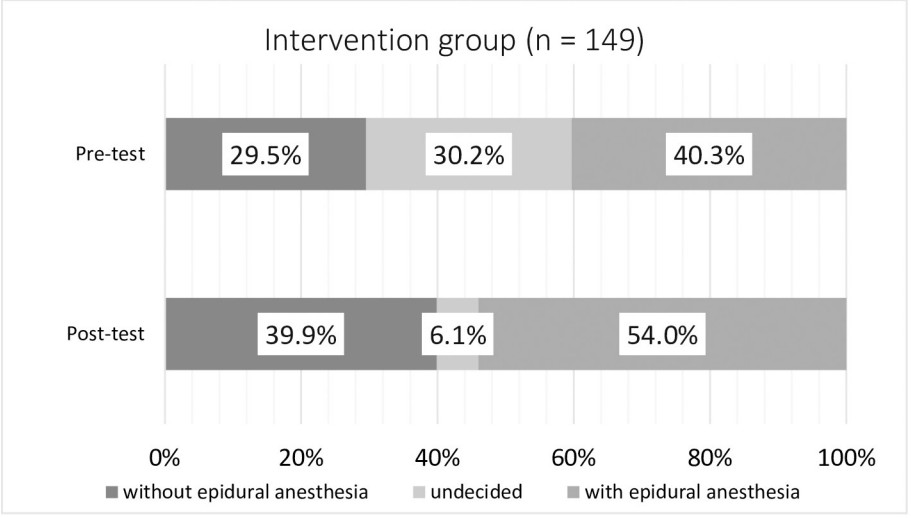

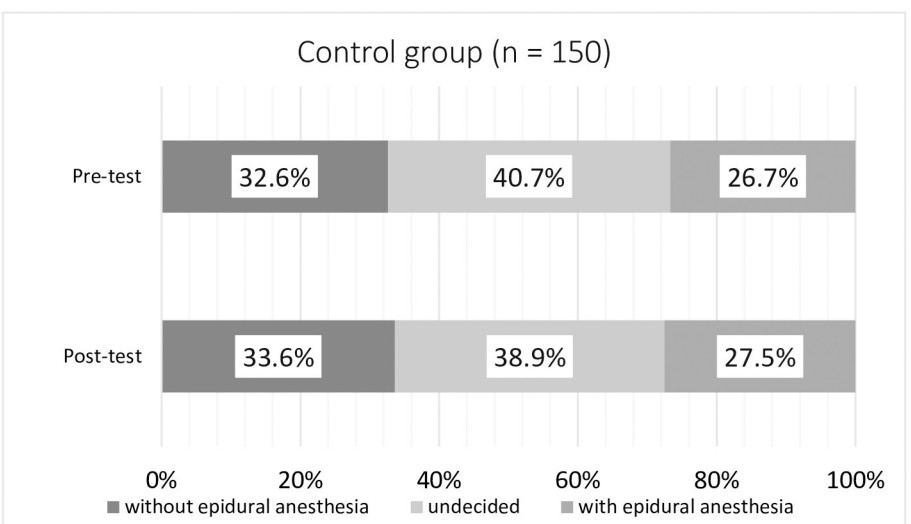

**Fig 3. Percentage of changes in the decision on using anesthesia: Without epidural anesthesia, undecided, or with epidural anesthesia.**

## Decisional conflict

Our study found that the mean decision conflict scores were significantly lower in the intervention group than in the control group. Our results were consistent with the results of the systematic review of Stacey et al. [10]. Their review integrated 105 studies on DAs from different disciplines. They used the DCS to assess decision conflict in the areas of cancer screenings, treatments for diabetes and tooth decay, and vaccinations in 63 out of the 105 studies. They found lower decision conflict scores in the intervention group which used DAs than in the control group which did not use DAs (MD -9.28, 95% CI: -12.2, -6.36).

The effects of DAs during the perinatal period have also been examined in 2 previous studies [11, 22]. One RCT study [22] of women with breech presentation found significantly lower decision conflict scores in the intervention group than in the control group. The other study [11] did not have the same findings (MD -0.99, 95% CI: -3.07, 1.07). The underlying reason for

the absence of a significant difference was presumed to be the number of options from which the participants could choose from. The study had 11 options and multiple options were available. It is possible that there was less conflict to begin with when there are many options to choose from. Other studies [10, 23] differed in that respondents can only select 1 from 2 options. In the present study, it was suggested that the effect of the DA was ascertained because the respondents could only choose 1 from 3 options: with epidural anesthesia, without epidural anesthesia, or undecided.

Comparing the scores of the subscale items of the DCS between the intervention and control groups in the present study, the mean score differences for the items Informed, Values of Clarity, Support, and Uncertainty were significantly lower in the intervention group than in the control group. Regarding the effects of DAs in the systematic review of Stacey et al. [10], it was reported that the participants in the intervention group felt more educated, could be more accurately recognized, could make better choices, and gains more clarity about its values than the participants in the control group. Therefore, it was suggested that obtaining the correct information might have helped the participants readily clarify their values and make their own choice.

## Satisfaction with decision-making, knowledge of epidural anesthesia, and choice for epidural anesthesia

In the present study, the mean score difference of Satisfaction with decision-making was higher in the intervention group than in the control group. In their RCT [11] of using DAs for assisting informed decision-making for labor analgesia, Raynes-Greenow et al. (2010) also found a significant increase in the score of decision-making satisfaction in the intervention group with a DA for pain relief compared with the control group. The present results were also consistent with a systematic review of DAs across a wide variety of situations [23]. We speculate that providing information on both benefits and risks would increase Satisfaction with decision-making.

The mean score difference of Knowledge of epidural anesthesia was also higher in the intervention group than in the control group. In their RCT study in 2007, Raynes-Greenow et al. [10] reported that the knowledge score increased in their study, and their results were similar to those of a systematic review of DAs across a wide variety of situations [10]. This implies that correct knowledge was acquired using the DA.

The proportion of pregnant women who had not decided on the use of analgesia during labor decreased in the intervention group compared with the control group. This was similar to the results of the systematic review of Stacey et al. (RR 0.64, 95% CI: 0.52, 0.79) [10]. In the systematic review of Lally et al. (2008) [24], pregnant women's knowledge during pregnancy could increase their confidence and lead to a better experience. For pregnant women who have not decided whether to use epidural anesthesia or not, DAs have been suggested to be useful as a tool for assisting decision-making. In 1994, Brown and Lumley [13] described the satisfaction of 790 Australian women with their care at birth within the first week of delivery. They reported that the lack of involvement in decision-making and the lack of adequate information had a significant impact on the dissatisfaction of the women at the time of their delivery. In a focus group study, the women were reportedly unable to describe the risks and benefits of labor analgesia even though they had considered themselves to be knowledgeable about the topic [25]. It was suggested that the right information and making your own decisions during pregnancy could lead to delivery satisfaction.

## Limitations

To our knowledge, this is the first study in Japan regarding the use of a DA for assisting informed decision-making whether to use epidural anesthesia or not, including preferences.

In this study, we used a reliable, previously confirmed, and valid scale (i.e., DCS) for the primary outcome, and a nonequivalent control group that controlled for internal validity [21, 26, 27]. A feature of a nonequivalent control group design is that both naturally occurring groups will be similar. A limitation in this study is that it is not known if the 2 groups were similar because of missing educational data, which was primarily from the control group. Also, we were not able to ascertain the knowledge gained by the participants from other sources of information (e.g., books, internet) between the pre-test and the post-test. It is important to state that the subjects of this study were limited to Japanese women, as access to analgesia and pain relief differs by country, culture, and medical system. As this study was not an RCT, bias in the results cannot be completely ruled out. A subsequent RCT is necessary to allow generalization of the results. In the future, we need to consider against selection bias in parity and previous surgery/delivery. An assessment as to how a rigorous RCT could be performed is underway as the DA is being planned to be uploaded and viewed on the web rather than in paper form.

## Conclusions

To our knowledge, this study is the first non-RCT to evaluate the effect of a DA on pregnant women's decision on using anesthesia for labor during vaginal delivery, namely, with epidural anesthesia, without epidural anesthesia, or undecided. The women who used a DA had a significantly lower DCS score, higher satisfaction and knowledge scores, and lower indecision rate than the women who used only a pamphlet as the source of childbirth information.

The goals of using DAs are to inform pregnant women about the benefits and risks of each delivery option and to ensure that they are congruent with their own values. DAs could therefore be useful for pregnant women who are undecided in using epidural anesthesia for labor during vaginal delivery.

## Supporting information

**S1 Dataset.**
(XLSX)

## Acknowledgments

We greatly appreciate **Dr. Edward Barroga** (http://orcid.org/0000-0002-8920-2607), Medical Editor and Professor of Academic Writing at St. Luke's International University for his editorial review and guidance in writing the article.

## Author Contributions

**Conceptualization:** Eri Shishido, Wakako Osaka, Ayame Henna.

**Data curation:** Yuko Motomura.

**Formal analysis:** Yuko Motomura, Shigeko Horiuchi.

**Funding acquisition:** Shigeko Horiuchi.

**Methodology:** Eri Shishido, Ayame Henna.

**Project administration:** Eri Shishido.

**Supervision:** Wakako Osaka, Shigeko Horiuchi.

**Writing – original draft:** Eri Shishido, Wakako Osaka, Yuko Motomura, Shigeko Horiuchi.

**Writing – review & editing:** Eri Shishido, Ayame Henna, Shigeko Horiuchi.

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
