## [Decision Letter · Decision Letter 0]

28 Jul 2020

PONE-D-20-18281

Effect of a decision aid on the choice of pregnant women whether to have epidural anesthesia or not during labor

PLOS ONE

Dear Dr. Shishido,

Thank you for submitting your manuscript to PLOS ONE. After careful consideration, we feel that it has merit but does not fully meet PLOS ONE’s publication criteria as it currently stands. Therefore, we invite you to submit a revised version of the manuscript that addresses the points raised during the review process.

We look forward to receiving your revised manuscript.

Kind regards,

David Desseauve, MD, MPH, PhD

Academic Editor

PLOS ONE

Journal Requirements:

"This study was supported by grants from the Japan Society for the Promotion of Science

(P1-Shigeko Horiuchi 17H01613) and (P2-Naoko Arimori 17H04427)

(https://www.jsps.go.jp/english/e-grants/grants01.html)."

"No"

3. Please include your tables as part of your main manuscript and remove the individual files. Please note that supplementary tables (should remain/ be uploaded) as separate "supporting information" files.

4. Please upload a copy of Supporting Information Files which you refer to in your text on page 22.

Reviewers' comments:

Reviewer's Responses to Questions

**Comments to the Author**

1. Is the manuscript technically sound, and do the data support the conclusions?

Reviewer #1: Yes

Reviewer #2: Yes

Reviewer #3: Yes

Reviewer #4: Yes

2. Has the statistical analysis been performed appropriately and rigorously? 

Reviewer #1: Yes

Reviewer #2: Yes

Reviewer #3: I Don't Know

Reviewer #4: Yes

3. Have the authors made all data underlying the findings in their manuscript fully available?

Reviewer #1: Yes

Reviewer #2: Yes

Reviewer #3: Yes

Reviewer #4: Yes

4. Is the manuscript presented in an intelligible fashion and written in standard English?

Reviewer #1: Yes

Reviewer #2: Yes

Reviewer #3: No

Reviewer #4: Yes

5. Review Comments to the Author

Reviewer #1: Thanks to editorial committee for the opportunity to review this work. The paper aim to investigate the effect of a decision aid tool for women in order to help them to make a decision between planned epidural analgesia and planned no epidural analgesia. The authors hypothesize that a specific decision aid tool may reduce the part of prenatal undecided women. The paper is well written original. There are some limitations that should be addressed in a revised version. It is unclear why the authors did not planned a randomized trial, a power calculation is missing, the assignation procedure into a group is unclear, there is no evaluation of postnatal satisfaction about pain management.

You will find my detailed comments below

Detailed comments

Abstract

- The first paragraph is wordy “decision aid on pregnant women’s decisions….”

- The objective reported here is non informative, we don’t know what is the main analysis

- Please report, at least, the number of included women and indicate how women were assigned into the intervention group or not

- Considering that the main objective and the main judgement criterion are not clearly defined, conclusions appears not interpretable

Introduction

- Consider providing data explaining why such a low rate of epidural analgesia compared to other countries like France with more than 50% of woman benefiting from epidural analgesia.

- The review of literature should be condensed for reducing the volume of the introduction part which is too long

- Lines 70-74 : what about information delivered by anesthesiologists ?

- In the last paragraph, the main objective and related and judgement criterion remains unclear to me. This should be clarified. Is that as follows: investigating the effect of a DA on prenatal women’s decision about epidural analgesia use; with / without / undecided. If it is you may consider formulate an hypothesis, meaning that you hypothesis that DA increase the rate prenatal epidural wishes.

Methods

- Justify why not implement a randomized trial

- It seems that you included both nulliparous and parous women. Don’t you think that it could biases your analysis?

- There is no justification for including 300 women. Your research question and the design requires an a priori power calculation. Formulate a hypothesis and, based on it, a power calculation. This approach is strongly limited by the fact that it is an a posteriori calculation and may be discussed below.

- How women were assigned to a group? This must be clearly reported

- Finally you compared a 22 pages pamphlet versus a 11 pages pamphlet…. By reading this it seems to me that you compared the effect of two information pamphlet about analgesia during labor on women’s decision and not really a decision aid tool versus not…I mean here that both might be considered as decision aid tools with one more detailed than the other.

- Please clearly expose your main objective being concordant with the primary outcome definition

Results

- The difference about educational level between your 2 groups affect the validity of the results. We need a justification for not implementing a randomized trial and a more detailed description of the procedure allocation to a group

Discussion

- In my opinion the key message of the paper is the reduction of undecided women and this point deserve to more highlighted in your paper

- I think that the most important criteria would have been “postnatal women’s satisfaction about pain management”. Discuss the fact that the primary outcome may lack of clinical significance. Ok, it is great women are less undecided with your DA but if the postnatal satisfaction about pain management is extremely low ….the impact of your DA on women’s health is none. If a randomized trial is implemented, this point should be addressed.

Reviewer #2: Although the subject studied depends on the country's practices, this study is original and conducted with a rigorous method. It is based on the use of several validated scales clearly presented and explained. The results can be extrapolated to other choice situations in perinatal care. Limitations are clearly stated.

The study emphasizes the importance of the quality of information and the tools to improve it.

Reviewer #3: Cf document attached

Objectives

This study aimed to evaluate the effect of a decision aid (DA) on pregnant women’s

decision on using anesthesia for labor during vaginal delivery, namely, with epidural

anesthesia, without epidural anesthesia, or undecided.

Methods

In this non-randomized controlled trial, low-risk pregnant women in a Japanese urban

hospital were recruited and assigned to 2 groups based on the source of childbirth

information, namely, a DA group and a pamphlet group. Pre-test and post-test scores

in each group were evaluated using the Decision Conflict Scale (DCS) for the primary

outcome, knowledge of epidural anesthesia and satisfaction with decision making for

the secondary outcomes, and the decision of anesthesia usage, namely, with epidural

anesthesia, without epidural anesthesia, or undecided. The Institutional Review Board

of St. Luke’s International University, Tokyo, Japan approved the study protocol

(18–A069).

Results

The women in the DA group had significantly lower DCS score and higher satisfaction

rate than the women in the pamphlet group. The knowledge of epidural anesthesia

score of the women who used the DA was significantly higher than that of the women

who used the pamphlet. The women in the DA group showed a significantly lower

indecision rate than those in the pamphlet group.

Conclusion

Women who used the DA had significantly lower DCS score, higher satisfaction rate,

higher knowledge score, and lower indecision rate than women who used only the

pamphlet as the source of childbirth information.This study suggests that a DA can be

useful for women in deciding whether to use epidural anesthesia or not for labor during

vaginal delivery.

Reviewer #4: General comment:

The article shows data from a single-center non-randomized controlled trial taking place in Japan. The topic of this study is of interest. As described by the authors, few publications focus on decision aids on pregnant women although these are very interesting tools to promote shared-decision making process. Here are my comments throughout the manuscript:

Abstract

Please detail the number of women included in each group as well as the difference in means for each outcome.

Introduction

The introduction is globally clear. Some points could be clarified:

Compared to other developed country, the rate of epidural analgesia seems very low in Japan. Could you explain a little more these low rates and the reason for the low use of epidural analgesia (cultural? economic? organizational?). Precise in the method section the rate of epidural analgesia in your birth centre.

It seems that you consider negatively being ‘undecided’ or in ‘decision conflict’ for pregnant women regarding their wish to have or not epidural analgesia. Is there any data to support this point (lower satisfaction, increased anxiety, poorer outcomes? Etc.), in the context of epidural analgesia or more globally in studies dealing with shared-decision making? It makes sense that, even with the best level information and knowledge, some women, especially primiparous who have never experienced such level of pain, remained undecided about analgesia until delivery, because they cannot anticipate their wish or not have analgesia.

Methods

Please be more precise, if possible, when describing how women receive information about epidural analgesia: who give the pamphlet or DA (midwife or anaesthetist) and is the document is completed by an oral information? Is it possible that the information given (apart from the document) was different between the 2 periods?

To get an idea of the reading effort for pregnant women, you may detail le time required to read each document in Japanese (pamphlet and DA).

Do the DA describe alternative pain management methods?

Why did you choose to measure DSC and satisfaction at one week and not at or just after delivery?

Statistical analysis

Please precise how you define the study sample.

It is not clear in the statistical analysis part that the outcomes will be assessed by differences in score means. It should be specified.

Before performing Student t test, have you checked that hypothesizing a normal distribution seemed correct? Please consider justifying that somewhere.

Results and figure

A Flowchart should be added with the exact number of eligible women during the period and and the number of women non included with the reason. It could also be interesting, if possible, to present as additional data the comparison of eligible women included versus non included to document selection bias.

In addition to the differences of means, you should present the means score for the pre-test for each group to see if there was baseline difference. For example, how to explain that there seems to be a difference in the rate of indecision (Figure 1), which was more important at baseline in the control versus intervention group? Can we assume that the discourse of caregivers changed in intervention period?

Presenting these means would also allow us to know the mean level of knowledge, satisfaction and DCS. If the difference is significant but the overall level is low, the tools remain insufficient.

Figure 1: you should detail the percentage rather than the number for clarity.

Discussion

Please discuss why you couldn’t do a randomized controlled trial in the limit section of your manuscript. It could have prevented from comparability bias.

Have you checked that your results were not affected by the difference in educational level between control and intervention group by doing sensitivity analysis among high education level women? It may help you to discuss the potential selection bias.

6. PLOS authors have the option to publish the peer review history of their article (what does this mean?). If published, this will include your full peer review and any attached files.

Reviewer #1: No

Reviewer #2: No

Reviewer #3: No

Reviewer #4: No

---

## [Author Response · Author response to Decision Letter 0]

12 Oct 2020

Subject: Submission of revised manuscript (PONE-D-20-18281) entitled, “Effect of a decision aid on the choice of pregnant women whether to have epidural anesthesia or not during labor” by Shishido et al.

Dear Dr. Desseauve,

Thank you for your careful review and valuable suggestions regarding our manuscript. We greatly appreciate the constructive comments by the reviewers which have helped us to considerably improve our paper.

We have thoroughly revised our manuscript in accordance with all the reviewers’ comments. All revisions and newly added text in the revised manuscript are indicated by track changes in accordance with the resubmission instructions. We have also provided below our point-by-point responses to each of the comments raised.

The revised manuscript has also been comprehensively reviewed and edited for English language by a professional native English-speaking medical and nursing science editor to meet the language standards required by leading English-language publications. The editing did not involve any alteration of the research content or the authors’ intentions.

We hope that we have satisfactorily answered all the concerns of the reviewers, and that the revised manuscript is now acceptable for publication in PLOS ONE.

Thank you for considering our revised manuscript. We look forward to hearing from you at your earliest convenience.

Sincerely yours,

Eri Shishido, PhD, MN, CNM

St. Luke’s International University

10-1 Akashi-cho, Chuo-ku, Tokyo104-0044, Japan

Phone: +81-3-3543-6391

Fax: +81-3-5565-1626

E-mail: 17DN003@slcn.ac.jp

---

## [Decision Letter · Decision Letter 1]

2 Nov 2020

Effect of a decision aid on the choice of pregnant women whether to have epidural anesthesia or not during labor

PONE-D-20-18281R1

Dear Dr. Shishido,

We’re pleased to inform you that your manuscript has been judged scientifically suitable for publication and will be formally accepted for publication once it meets all outstanding technical requirements.

Kind regards,

David Desseauve, MD, MPH, PhD

Academic Editor

PLOS ONE

Additional Editor Comments (optional):

Reviewers' comments:

Reviewer's Responses to Questions

**Comments to the Author**

1. If the authors have adequately addressed your comments raised in a previous round of review and you feel that this manuscript is now acceptable for publication, you may indicate that here to bypass the “Comments to the Author” section, enter your conflict of interest statement in the “Confidential to Editor” section, and submit your "Accept" recommendation.

Reviewer #1: All comments have been addressed

Reviewer #3: All comments have been addressed

2. Is the manuscript technically sound, and do the data support the conclusions?

Reviewer #1: Yes

Reviewer #3: Yes

3. Has the statistical analysis been performed appropriately and rigorously? 

Reviewer #1: Yes

Reviewer #3: Yes

4. Have the authors made all data underlying the findings in their manuscript fully available?

Reviewer #1: Yes

Reviewer #3: Yes

5. Is the manuscript presented in an intelligible fashion and written in standard English?

Reviewer #1: Yes

Reviewer #3: Yes

6. Review Comments to the Author

Reviewer #1: Thank your for the opportunity to review this revised version of the manuscript.

In my opinion the authors performed a great review of their initial manuscript.

Reviewer #3: Dear Editor,

Thank you for giving me the opportunity to assess the resubmission 1 of this original manuscript intended for publication in PlosOne.

The authors aim through their study to compare the effect of a decision aid (DA) on pregnant women’s decision on using epidural anesthesia or not for labor during vaginal delivery.

They have thoroughly revised their manuscript in accordance with all of my comments.

The point-by-point response is self-explanatory.

They have thus extensively and with great quality (Bravo) answered all the concerns of the review,

I hereby consider the manuscript acceptable for publication in PLOS ONE.

7. PLOS authors have the option to publish the peer review history of their article (what does this mean?). If published, this will include your full peer review and any attached files.

Reviewer #1: No

Reviewer #3: No

---

## [Editor Report · Acceptance letter]

4 Nov 2020

PONE-D-20-18281R1 

Effect of a decision aid on the choice of pregnant women whether to have epidural anesthesia or not during labor 

Dear Dr. Shishido:

I'm pleased to inform you that your manuscript has been deemed suitable for publication in PLOS ONE. Congratulations! Your manuscript is now with our production department. 

Kind regards, 

on behalf of

Dr. David Desseauve 

Academic Editor

PLOS ONE